# Biomarkers and Lung Cancer Early Detection: State of the Art

**DOI:** 10.3390/cancers13153919

**Published:** 2021-08-03

**Authors:** Elisa Dama, Tommaso Colangelo, Emanuela Fina, Marco Cremonesi, Marinos Kallikourdis, Giulia Veronesi, Fabrizio Bianchi

**Affiliations:** 1Cancer Biomarkers Unit, Fondazione IRCCS Casa Sollievo della Sofferenza, 71013 San Giovanni Rotondo, Italy; e.dama@operapadrepio.it (E.D.); t.colangelo@operapadrepio.it (T.C.); 2Humanitas Research Center, IRCCS Humanitas Research Hospital, 20089 Rozzano, Milan, Italy; emanuela.fina@humanitasresearch.it; 3Adaptive Immunity Laboratory, IRCCS Humanitas Research Hospital, 20089 Rozzano, Milan, Italy; marco.cremonesi@humanitasresearch.it (M.C.); Marinos.Kallikourdis@humanitasresearch.it (M.K.); 4Department of Biomedical Sciences, Humanitas University, 20072 Pieve Emanuele, Italy; 5Division of Thoracic Surgery, IRCCS San Raffaele Scientific Institute, 20132 Milan, Italy; veronesi.giulia@hsr.it

**Keywords:** lung cancer, early diagnosis, biomarkers, liquid biopsy

## Abstract

**Simple Summary:**

Lung cancer is the leading cause of cancer death worldwide. Detecting lung malignancies promptly is essential for any anticancer treatment to reduce mortality and morbidity, especially in high-risk individuals. The use of liquid biopsy to detect circulating biomarkers such as RNA, microRNA, DNA, proteins, autoantibodies in the blood, as well as circulating tumor cells (CTCs), can substantially change the way we manage lung cancer patients by improving disease stratification using intrinsic molecular characteristics, identification of therapeutic targets and monitoring molecular residual disease. Here, we made an update on recent developments in liquid biopsy-based biomarkers for lung cancer early diagnosis, and we propose guidelines for an accurate study design, execution, and data interpretation for biomarker development.

**Abstract:**

Lung cancer burden is increasing, with 2 million deaths/year worldwide. Current limitations in early detection impede lung cancer diagnosis when the disease is still localized and thus more curable by surgery or multimodality treatment. Liquid biopsy is emerging as an important tool for lung cancer early detection and for monitoring therapy response. Here, we reviewed recent advances in liquid biopsy for early diagnosis of lung cancer. We summarized DNA- or RNA-based biomarkers, proteins, autoantibodies circulating in the blood, as well as circulating tumor cells (CTCs), and compared the most promising studies in terms of biomarkers prediction performance. While we observed an overall good performance for the proposed biomarkers, we noticed some critical aspects which may complicate the successful translation of these biomarkers into the clinical setting. We, therefore, proposed a roadmap for successful development of lung cancer biomarkers during the discovery, prioritization, and clinical validation phase. The integration of innovative minimally invasive biomarkers in screening programs is highly demanded to augment lung cancer early detection.

## 1. Introduction

Lung cancer is an aggressive disease accounting for ~380,000 deaths/year only in Europe (WHO; http://gco.iarc.fr; accessed on 21 April 2021) and ~2 million deaths/year worldwide. With the COVID-19 pandemic, these rates are unfortunately expected to rise, mainly due to delays in screening, hospitalizations and therapies, which will cause a stage-shift for newly diagnosed lung tumors [1,2,3].

Detecting lung malignancies promptly is essential for any anticancer treatment to reduce mortality and morbidity, especially in high-risk individuals [4]. The US National Lung Screening Trial (NLST) and other non-randomized trials [5] demonstrated that Low-Dose Computed Tomography (LDCT) screening can reduce mortality (~20%). Recently, the European NELSON trial has observed a lung cancer mortality reduction of ~25% at 10 years and up to ~30% at 10 years [6]. The drawback of LDCT screening is the presence of uncertainties about high costs, risk of radiation exposure, and false positives observed in the screening population [7], which may obstacle a fully safe large scale implementation of the LDCT screening for lung cancer in Europe [8]. The false-positive rate is particularly problematic, as suspicious nodules may require invasive investigations, causing unnecessary morbidity and reduced acceptance of screening among at-risk individuals. Therefore, the integration of LDCT screening with innovative cancer biomarkers analyzable through minimally invasive approaches aimed to increase screening accuracy is highly demanded. Several pre-clinical studies have suggested that circulating molecules such as microRNA, DNA, proteins, autoantibodies in the blood, as well as circulating tumor cells (CTCs), could be potentially useful to diagnose lung cancer and increase screening accuracy [9,10,11,12]. In addition, some studies in actual lung cancer screening cohorts confirmed the diagnostic validity of measuring blood biomarkers for lung cancer early detection [13,14,15]. Yet, pitfalls and caveats emerged during validation of some proposed biomarkers for lung cancer early detection once applied to independent cohorts/multicenter studies and/or actual lung cancer screening cohorts, which highlight the need to establish a roadmap to develop effective biomarkers.

We reviewed the literature for the most promising biomarkers and relevant technical issues, of which here we present a summary with the aim to propose guidelines for an accurate study design and execution, and data interpretation for biomarker development. We hope that these guidelines will aid further research and facilitate the translation of circulating biomarkers into clinical setting.

## 2. Lung Cancer Biomarkers

In the last 10 years, there has been a sharp rise in published studies on lung cancer diagnostic biomarkers, with over 544 papers published only in the last 5 years (Figure 1A). However, a sizable fraction of these works relies on a relatively small cohort of samples analyzed, without validation of biomarkers in independent cohorts and, more importantly, in lung cancer screening trials. Ideally, robust biomarker(s) should facilitate the selection of at-risk individuals independently of risk factors such as age and smoking habits, and/or provide pathological information about indeterminate pulmonary nodules (IPNs) to aid clinical decision making, and/or provide predictive/prognostic information. Here, we focused on the most promising minimally invasive, reproducible and extensively validated biomarkers assessed in prospective studies, including lung cancer screening trials.

### 2.1. DNA-Based Biomarkers

Circulating tumor DNA (ctDNA) was extensively investigated in the latest years due to recent technological advances in the field of next-generation sequencing (NGS). Indeed, the NGS technologies allow the analysis of custom panels of genes (i.e., targeted gene panels, TGP) at an affordable cost (~330€ per sample; [16]) and the detection of mutant alleles presenting with low frequency (<1%; [11,17,18]), which is mandatory when dealing with ctDNA, i.e., underrepresented among the more abundant cell-free DNA (cfDNA) of hematopoietic origin. Although ctDNA was shown to be effective in the diagnosis of advanced lung cancer, the use of ctDNA for detection of early stage lung tumors is suboptimal (with sensitivity ranging from ~50%, [11,19] to 15% in the case of stage I NSCLC; [20]), which can be ascribable to the rare amount of ctDNA present in blood samples of stage I disease patients; indeed, the low proliferation/metabolic rate, and/or dismal tumor angiogenesis, and/or lack of necrotic areas of these localized and tiny tumor lesions all contribute to a reduced ctDNA shedding, as recent observations have suggested [21].

Furthermore, commercial TGPs are usually designed to track druggable cancer driver mutations in more advanced cancer which, therefore, can be underrepresented in early stage disease, i.e., characterized by lower intra-tumor genetic heterogeneity [22,23,24]. Consequently, the chance to capture nucleotide variants in ctDNA of stage I is dismal. As an alternative, some groups applied the CAncer Personalized Profiling by deep Sequencing (CAPP-Seq) [11] in liquid biopsies to overcome the limited sensitivity of more standard approaches. CAPP-seq introduced a preliminary bioinformatics approach to select target genes containing regions recurrently mutated in the cancer of interest [11]. Despite significant results reached by applying such technology to track molecular residual disease (MRD) during lung cancer therapy [25], the application of CAPP-seq for diagnosis of early stage lung cancer still resulted in a suboptimal sensitivity (~50% [21]). Whole-exome (WES) or whole-genome (WGS) sequencing [26] of ctDNA, covering the entire set of known human genes in order to overcome limitations of TGP, have been also attempted [27]. However, it should be kept in mind that the larger the gene panels, the more difficult it is to obtain high sensitivity for mutation calling and to maintain affordable costs. The high level of ctDNA fragmentation (~100–150 bp in size; [27,28]) should also be considered when designing libraries for NGS. Other caveats in the detection of ctDNA are related to clonal hematopoiesis (CH), i.e., an age-dependent process determining the accumulation of somatic mutations in hematopoietic stem and progenitor cells ultimately leading to the clonal expansion of mutated hematopoietic cells; CH accounts for the non-tumor derived mutations detected from plasma [29]. Therefore, it is worth considering to sequence matched white blood cell (WBC) DNA and cfDNA to determine the tumor specific fraction of cfDNA mutations.

Beyond detecting ctDNA mutations, other groups described methylation profiling of cfDNA as a source of innovative minimally invasive cancer biomarkers. A global hypomethylation of DNA is usually observed in cancer cells, yet hypermethylated regions overlapping with CpG islands promoters of tumor suppressor genes were also discovered and exploited to detect ctDNA [30]. The analysis cfDNA using specific methylation signatures to estimate the ctDNA fraction was indeed showed to be a valuable approach for diagnostic and prognostic purposes in lung cancer [31,32]. In a recent large trial with a multi-cancer cohort of over 6000 participants, the methylation profile of ctDNA was found to be highly specific (~99.3%) and to reach an acceptable sensitivity of 67.3% in a set of 12 cancer types and including lung cancer. However, sensitivity dropped down when analyses were limited to early-stage disease (39%; <25% in lung cancer) [33], thus suggesting the need for further investigation of cfDNA methylation signatures in actual lung cancer screening trials for refinement and validation.

### 2.2. RNA-Based Biomarkers

Different circulating RNA species (microRNA, miRNA; piwi-interacting RNAs, piRNA; transfer RNAs, tRNA; small nucleolar RNAs, snoRNA; small nuclear RNAs, snRNA) were identified in the human serum [34]. Circulating microRNAs (c-miRNAs) are predominant in the literature, and their remarkable stability in harsh conditions and resistance to circulating RNAses [35] make them ideal candidates for developing lung cancer biomarkers. C-miRNAs are released by virtually all human cells by passive (e.g., in apoptotic bodies, complexed with AGO proteins) and active (e.g., in exosomes [36]/microvescicles) mechanisms [37], and can influence tissue homeostasis by a sort of paracrine signaling [37] or by triggering pathogenic mechanisms including neoplastic transformation and tumor progression [38,39]. Indeed, tumor cells, cancer-associated fibroblasts (CAFs) and blood cells were found to release miRNAs in the microenvironment which then enter into the bloodstream [37,40].

Therefore, monitoring miRNA species and relative quantities in the blood represents a valid strategy for early diagnosis of lung cancer. Few studies underwent an extensive validation of c-miRNA as minimally invasive biomarkers for lung cancer early detection (Table 1). Montani et al. validated [13] a serum 13 c-miRNAs signature (miR-Test) by using the qRT-PRC in high-risk individuals (*n* = 1115; >20 pack-year smoking history, aged >50 years) enrolled in an Italian LDCT screening trial (the COSMOS study), which showed a sensitivity of 0.78, a specificity of 0.75, and an AUC of 0.85. Likewise, Sozzi et al. [14] validated a 24 c-miRNA signature (the MSC classifier) by using the qRT-PCR in plasma samples of high-risk subjects enrolled in another Italian LDCT screening study (the bioMILD study; *n* = 939 participants), with a sensitivity of 0.87 and a specificity of 0.81. Wozniac et al. [41] analyzed plasma samples of 100 non-small cell lung cancer (NSCLC) patients (stage I–IIIA) and 100 healthy subjects, using the same qRT-PCR technology as the one used by the Italian studies, and identified another set of 24 miRNAs showing a predicted AUC of 0.78 when accounting for overfitting [41]. In Appendix A, we reported overlapping c-miRNAs in the various signatures identified by qRT-PCR.Notably, authors meta-analyzed the MSC classifier as well as another 34 c-miRNA signature identified by Bianchi et al. [9] (from which the miR-Test was derived) and reported an AUC of 0.70 and 0.78, respectively [41]. 

In multiethnic and multicentric studies on NSCLC patients and matched controls (lung cancer-free or with benign lung nodule individuals), Wang et al. [42] and Ying et al. [43], using the qRT-PCR, have identified two serum c-miRNA diagnostic signatures composed by 5 miRNAs each (miR-214 was commonly found; Appendix A). Other studies using different screening platforms, such as microarray analysis of serum samples [44] or whole-blood samples [45], have identified lung cancer diagnostic c-miRNA using large cohorts of clinically detected lung cancer patients (Table 1).

**Table 1 cancers-13-03919-t001:** List of studies reporting the development of c-miRNA-based biomarkers diagnostic for lung cancer.

Authors	PubMed ID	miRNA (*n*)	AUC	Sample Type	LDCT
Boeri et al. [46]	21300873	13	0.88	Plasma	Yes
Sozzi et al. [14]	24419137	24	- ^a^	Plasma	Yes
Bianchi et al. [9]	21744498	34	0.89	Serum	Yes
Montani et al. [13]	25794889	13	0.85	Serum	Yes
Wozniak et al. [41]	25965386	24	0.78 ^b^	Plasma	No
Shen et al. [47]	21864403	3	0.86	Plasma	No
Lin et al. [48]	28580707	3	0.87	Plasma	No
Chen et al. [49]	21557218	10	0.97	Serum	No
Wang et al. [42]	26629532	5	0.82	Serum	No
Ying et al. [43]	32943537	5	0.91–0.97	Serum	No
Zhu et al. [50]	27093275	4	0.97 ^c^	Serum	No
Nadal et al. [51]	26202143	4	0.99	Serum	No
Asakura et al. [44]	32193503	2	0.99	Serum	No
Fehlmann et al. [45]	32134442	15	- ^d^	Blood	No

The number of miRNA (*n*) in each diagnostic signature is reported together with the performance (AUC, i.e., area under curve) and the type of biospecimen where biomarkers were derived (Serum or Plasma). LDCT, studies which performed validation of biomarkers on actual LD-CT screening trials (Yes). ^a^ Sensitivity, 88% and a specificity of 80%; ^b^ Predicted performance when applied to independent samples. ^c^ miRNAs combined with carcinoembryonic antigen (CEA). ^d^ Sensitivity, 82.8%, and a specificity of 93.5%. PubMed identifiers (PubMed ID) are reported to allow retrieving cited publications.

Despite the proven validity of most of these c-miRNA signatures for early diagnosis of lung cancer, there are still limitations in their application in medical laboratories. Challenging issues related to sample processing and miRNA profiling, pre-analytical and analytical standardization as well as the considerable cost of sophisticated technologies, make the translation of such biomarkers from the bench to the bedside very complicated.

Later, we will further discuss some of these limitations with the aim to provide guidelines for biomarker profiling and translation to the clinic.

### 2.3. Protein-Based Biomarkers

The ability of tumor antigens [12] and tumor-associated autoantibodies (TAABs) [52] in body fluids to serve as potential biomarkers for lung cancer early detection has been investigated for years. In 2015, Doseeva et al. [53] showed that the combined use of tumor antigens (CEA, CA-125, and CYFRA 21-1) and autoantibodies (NY-ESO-1) was accurate enough (sensitivity, 77%; and specificity, 80%) for the early detection of NSCLC among high-risk individuals. Analysis of CEA and CA-125 among others protein biomarkers (i.e., CA19-9, PRL, HGF, OPN, MPO and TIMP-1) were also included in a multi-analyte blood test (CancerSEEK; [19]), which increased the sensitivity in tumor detection when combined with ctDNA mutation profiling [19].

A large number of studies, systematically reviewed by Yang and colleagues [54], showed that lung cancer patients produce antibodies recognizing self-antigens (i.e., TAAbs). These TAAbs were tested as potential biomarkers for lung cancer detection at different stages of tumor progression. Among TAAbs, the New York esophageal squamous cell carcinoma-1 (NY-ESO-1) autoantibodies appeared to be most promising for NSCLC detection alone or in combination with other TAAbs [54]. However, the diagnostic utility would be more evident if patients affected by *bona fide* autoimmune disease could also be included in the analysis, in order to test whether TAAbs are actually specific for lung cancer.

Recently, the detection and quantification of complement activation fragments in plasma samples from high-risk individuals who underwent LDCT screening were found to be a valid strategy to identify lung cancer biomarkers [15]. A simple diagnostic model based on the quantification of complement-derived fragment C4c and cancer antigens, i.e., 21.1 (CYFRA 21-1) and C-reactive protein (CRP), was able to discriminate between benign and malignant pulmonary nodules (AUC, 0.86), with a high specificity (92%) in a cohort of individuals enrolled in a CT-screening program. This was an important finding due to the considerable fraction (~24%; [5]) of false positive findings by LDCT at the baseline. Authors also showed that the model combined with clinical factors can be valuable in patients with indeterminate pulmonary nodules (IPNs) to decide for more effective therapeutic strategies [55].

### 2.4. Immune Serum Conversion as Biomarker for Lung Precancerous Lesions

Quantification of inflammation, via measurement of systemic levels of pro-inflammatory cytokines released by activated immune cells, showed a correlation between inflammation and a higher risk for lung cancer incidence in smokers [56,57]. On the other hand, extensive independent analysis of cohorts of non-smokers confirmed the association between sustained inflammation and a higher risk of developing lung cancer [58,59,60,61,62,63,64,65,66,67,68]. In this sense, pro-inflammatory immune activity, which is reflected in the level of circulating cytokines, may be a contributing factor to tumorigenesis in the lung.

The immune system affects not only the tumorigenesis, but also the progression of the disease [69,70,71]. Thus, whilst research efforts have focused on inflammatory mediators for their potential roles as risk factors for lung cancer in healthy individuals, in parallel, inflammatory mediators have also been assessed for their role in tumor progression in patients with established tumors. Even early stage premalignant lesions are highly infiltrated by immune cells, suggesting that the immune system may affect the transition to malignant lesions [72]. Thus, inflammatory cytokines could drive the progression to malignancy. To date, a detailed and systematic characterization of circulating inflammatory cytokines in patients bearing premalignant lesions in the lungs is still largely missing.

Interestingly, in line with the hypothesis that chronic inflammation is detrimental during carcinogenesis and cancer progression, Ridker and colleagues have recently demonstrated that atherosclerotic patients treated systemically with canakinumab, an antibody inhibiting pro-inflammatory cytokine IL-1β, are protected from lung cancer development, most likely due to the reduction of pro-tumoral inflammation [73]. This seminal clinical finding further highlights how circulating immune mediators may be pivotal for lung cancer progression.

### 2.5. Circulating Tumor Cells (CTCs) for Lung Cancer Screening

In 2014, a ground-breaking paper showed that, by using a size-based enrichment technology (ISET^®^, Isolation by Size of Epithelial/Tumor cells), it was possible to detect cells with morphological features of malignancy (i.e., circulating tumor cells, CTCs) in blood samples of patients suffering of chronic obstructive pulmonary disease (COPD) [74]. The presence of CTCs was shown to anticipate the radiological diagnosis of stage I NSCLC [74], thus leading to an increasing interest around the diagnostic role of CTCs and their implementation as a possible biomarker in lung cancer screening programs.

CTCs can be defined as tumor cells in transit in the circulatory system. They originate from primary and secondary tumor sites and are endowed with the molecular features needed to overcome some of the numerous and challenging steps of the metastatic cascade, including intravasation, survival in the blood microenvironment and dissemination to distant organs [75,76]. CTCs are rare events, mixed with a huge number of other cell types, mainly erythrocytes (3.5–7 billion/mL) and leukocytes (4–11 million/mL), and occurring at a variable frequency, even less than 1 cell per milliliter of peripheral venous blood depending on the tumor type and stage [77,78].

CTC detection for lung cancer diagnosis was found to be promising in initial and explorative studies by Hofman and colleagues [10,74]. The same research group then launched a large multicenter prospective French trial (AIR study, NCT02500693), which enrolled a cohort of 614 high-risk subjects according to the NLST-UPSTF criteria (aged 55–74 years, 30 or more pack-year smoking history; current smokers or heavy smokers having quit in the last 15 years) in order to assess the diagnostic accuracy of CTCs detected by the ISET^®^ technology. However, the sensitivity of CTC analysis in detecting 19 lung cancers found at first low-dose computed tomography (LDCT) scan was low, i.e., ~26% [79].

Encouraging results in terms of detection rate were recently obtained using a 4-color FISH test (Table 2) performed on the peripheral blood mononuclear cell (PBMC) fraction isolated by density gradient centrifugation. Through this technique, it was possible to detect cells with at least 2 polysomies or gains in 4 loci involved in the NSCLC tumorigenesis or prognosis (i.e., at 10q22.3, 3p22.1, 3q29 loci, or at chromosome 10 centromere) in 89% of 107 patients with ≤30 mm diameter pulmonary nodules. Contrariwise, none of the 100 lung cancer-free control cases were scored positive when the cut-off value was ≥3 cells with genome abnormalities. Overall, sensitivity was 88.8%, specificity 100%, and accuracy 94.2% [80]. Although the frequency and number of PBMCs with aneuploidy was higher in patients compared to controls, both the validity of a cut-off value of at least 3 cells with aneuploidy to call as CTC-positive a lung cancer patient and the significance of the presence of a maximum of 2 cells with aneuploidy in individuals at high risk for lung cancer should be confirmed in further case series. However, this paper suggests that looking at the entire PBMC population, rather than selecting specific subsets of cells, and using DNA-based detection techniques could considerably augment test sensitivity and specificity. In another work the introduction of alternative protein markers besides cytokeratins (CKs), such as the glycolysis enzyme hexokinase 2 (HK2), increased the detection of CTCs in a cohort of 18 stage III lung adenocarcinoma patients without clinical evidence of distant metastases from 39% when considering CK^pos^CD45^neg^ to 61% when considering HK2^high^CD45^neg^ cell subsets [81]. This suggests that using epithelial markers alone may not be sufficient to detect CTCs in non-metastatic setting, and that by adding other markers such as metabolic gene expression analysis can improve lung cancer diagnostic accuracy.

Compared to cell-free circulating biomarkers, circulating cells represent an ideal and promising systemic ‘surrogate’ of a tissue as they offer the opportunity to investigate the entire cell at morphological, protein, RNA and DNA level, and to develop experimental models for functional studies. However, the analysis of CTCs in blood samples requires the enrollment of trained personnel and the acquisition of dedicated technologies to enrich blood samples and detect target cells unambiguously. Results of studies in the diagnostic and preoperative setting demonstrate that the accuracy and clinical validity of each kind of technical approach for CTC analysis is still variable and has to be carefully assessed and confirmed in large multicenter and validation trials.

## 3. A Roadmap to the Successful Development of Blood-Based Biomarkers for Lung Cancer Early Detection

The bottleneck for the successful translation of biomarkers to the clinical use generally lies in the suboptimal standardization in each step of the biomarker pipeline, including discovery, prioritization, and clinical validation. We prepared a summary of the main issues and the best practices in biomarker development (Figure 1B). The first fundamental step in biomarker discovery is establishing a high-quality design which includes making explicit hypotheses on the potential application/integration into current recommended screening programs as well as adopting enrollment protocols with clear inclusion and exclusion criteria for patients and controls. Moreover, heterogeneity (epidemiological, biological and molecular) needs to be considered as the driver for adequate sample size to fulfill the best design. Indeed, published studies often lack acceptable sample size with respect to the numerous phenotypic features that should be considered to widely represent the screening population [88], and the number of variables that should be analyzed to deconvolute the high level of genetic heterogeneity of lung cancer. To limit self-selection bias, instead of convenience selection of subjects (based on easy availability of the sample) [89], control populations should be identified based on matching criteria with the patients’ cohort, and extensively represent the actual incidence and prevalence of lung cancer in the screening population.

In the absence of standards for handling specimens (collection, storage and processing) and controls for pre-analytical factors, randomization and blinding should be applied to reduce bias from the experimental analysis. Indeed, quality and reproducibility of biomarkers can be influenced by uncontrolled pre-analytical conditions (i.e., fasting, lipemia, partial hemolysis [90]) and by sample collection bias, especially when the biomarker is labile or sensitive to temperature fluctuation or handling conditions (i.e., type of collection tubes, centrifugation steps, long-term or short-term storage, freeze/thaw cycles; [91,92]). We therefore suggest performing initial pilot experiments to measure the stability of circulating biomarkers, i.e.: (i) by testing different samples collection strategies, using different collection tubes for serum or plasma collection [93,94,95,96]; (ii) quantifying how much hemolysis (partial or hidden) can influence biomarker concentration [97,98], (iii) checking if analyte concentration is influenced by fasting status [90], and (iv) testing if different storage conditions (short-term vs. long-term; +4 or −20/−80 °C or liquid nitrogen) can alter biomarker quantity and quality [90]. After such analyses, a standard operating procedure (SOP) for sample collection and handling should be defined and rigorously applied to the specific biomarkers screening study.

Nowadays, high-throughput data allow the identification of many biomarkers acting jointly on the risk of lung cancer; these markers can be easily combined in a single multivariable statistical model; moreover, to avoid the resulting possible overfitting (i.e., capturing noise instead of the true underlying data structure), machine learning approaches with sample-splitting or cross-validation should be considered [99]. The performance of a new biomarker for the early detection of cancer is easily measured by true-positive and false-positive rates, and summarized through receiver operating characteristic curves (ROC). However, the “average” performance is often presented in the literature, with ROC calculated across all study subjects, while subgroup and/or multivariable analysis should better reveal the utility of biomarker testing in specific groups (i.e., tumor stages, nodule density, histotypes).

Exploration of biomarkers’ performance in subgroups could also help with ranking the selected candidates for clinical relevance. Moreover, when a new biomarker study is published, only limited discussion on the biological function of the candidates is reported, and assay/platform reproducibility and standardization are frequently lacking (see below). In our experience, an in-depth analysis of technical and biological variables which might have an impact on the detection and quantification of selected biomarkers should also be performed. For example, uncontrolled environmental conditions during sample processing could influence the quantification of biomarkers of interest. Marzi et al. [90] showed, by using an automated purification system based on spin columns for nucleic acid purification, that efficiency in miRNA extraction was inversely proportional to temperature increase during daily runs. Similar findings were also described by other research groups [100].

In the case of analysis of multiple biomarkers (e.g., DNA, RNA and protein), the collected samples (whole blood, plasma, serum) can be split in several aliquots which can be differently prioritized for processing based on stability of the biomarkers of interest; in case of RNA, which is more liable, the relevant sample aliquot can be processed immediately while other aliquots (for other biomarker types) can be processed subsequently. Likewise, the use of different extraction kits with or without additional centrifugation steps could affect quantities and species of the biomarkers of interest. Cheng et al. [101] showed that plasma samples can be contaminated by residual platelets, which impact most miRNA measurements (~70%), therefore authors suggested to add pre- or post-storage centrifugation steps in order to remove residual platelet contamination. Furthermore, miRNA quantities may vary depending on the kit used for extraction [102,103].

To keep track of the impact of these pre-analytical and analytical variables, we strongly recommend using endogenous and exogenous controls. In circulating miRNA, biomarker analysis measuring both endogenous controls (e.g., RNU6, RNU44, miR-16 [104]) and exogenous controls, e.g., synthetic miRNAs from other organisms (ath-miR-159a and/or cel-miR-39), allows monitoring sample degradation, extraction efficiency and performance of miRNA detection by using different screening platforms (e.g., qRT-PCR, ddPCR, microarray, NGS).

Lastly, the analytical translation in a clinically applicable platform and validation in a large prospective trial are both needed to complete validation of candidate biomarkers. Industrial and clinical partners could facilitate these phases, providing funding supports and know-how in large-scale test production, regulatory affairs and commercialization [88]. A major issue in the validation of biomarkers for lung cancer early detection is to prove its benefit in the context of screening programs, where lead- and length-time biases and overdiagnosis are peculiar. Therefore, the choice of the end-point is essential and, although biases could occur in interpreting causes of death, lung-cancer mortality reduction should represent the primary endpoint [99], then followed by the evaluation of overall mortality.

## 4. Overview of Platforms for Circulating Biomarkers Detection: A Focus on c-miRNA Detection

The performance of different screening platforms available in terms of sensitivity, specificity and reproducibility, as well as relative costs of analysis should also be considered in advance before starting biomarkers profiling. As previously described, c-miRNAs are the most discussed in the literature as promising biomarkers for lung cancer early diagnosis. Besides the several pre-analytical and analytical factors, which can impinge on the biomarker reliability as we previously discussed, some considerations should be made on the impact on the accuracy of c-miRNA biomarkers when using different experimental platforms and technologies for biomarkers detection.

To quantify c-miRNA expression, a variety of platforms have been developed so far, mainly based on quantitative PCR (qRT-PCR), microarray, or next-generation sequencing (NGS) technology. Recently, the efficiency and concordance of different miRNA profiling platforms were assessed [105,106,107,108]. In 2014, Mestdagh et al. [105] analyzed the expression level of 196 common miRNAs measured by 12 different application platforms to provide a sort of “miRNA quality control (miRQC)” analysis. They performed experiments with high and low RNA input amounts and organized output measurements into four groups to represent the various testing questions, i.e.: reproducibility, specificity, sensitivity, and accuracy [105]. Similar qRT-PCR platforms showed a different performance in terms of reproducibility and specificity [105]. Sensitivity, on the other hand, is very much technology-related, with qRT-PCR platforms (i.e., TaqMan Cards PreAmp; ThermoFisher) being superior to hybridization- (i.e., microarray) and sequencing-based platforms. Furthermore, the hybridization platforms displayed higher specificity, but lower detection rates compared to most of the qRT-PCR and sequencing platforms [105]. Overall, the authors reported that sensitivity and specificity have a deep and important inverse relationship [105].

Next-generation technologies are now also available for miRNA profiling. For example, Small RNA sequencing (RNA-Seq), in particular, was reported to be superior for discovery studies, but less useful for high-throughput or fast turnaround applications [105]. Furthermore, when various RNA isolation and library preparation protocols are used, the reproducibility of small RNA-seq is significantly and negatively affected [106,109]. Recently, Godoy et al. [107] evaluated a small RNA-seq method optimized for low-input samples [106,110,111] (i.e., liquid biopsy) to three relatively novel platforms, i.e., (i) the HTG Molecular’s EdgeSeq miRNA Whole Tran-scriptome Assay (EdgeSeq), (ii) the Abcam’s FirePlex (FirePlex), and (iii) NanoString’s nCounter (nCounter). These three platforms were selected for their rapid turnaround time and ease of use, properties that are attractive for biomarker assays. The authors used pools of synthetic RNA oligonucleotides and standardized extracellular RNA human plasma samples to assess reproducibility, bias, specificity, sensitivity, and accuracy. Briefly, the authors concluded that: (i) small RNA-seq was the most accurate, sensitive and specific method with an AUC of 0.99 for miRNA detection, which was superior to EdgeSeq (AUC = 0.97), nCounter (AUC = 0.94) or FirePlex (AUC = 0.81); (ii) EdgeSeq was the most reproducible and had the least detection bias; and (iii) nCounter was less sensitive than small RNA-seq, EdgeSeq, and FirePlex. Recently, Hong LZ et al. [108] performed a systematic evaluation of multiple qPCR platforms (MiRXES ID3EAL, Qiagen miScript, TaqMan Cards preAMP, Exiqon LNA), nCounter technology (NanoString) and miRNA-Seq for microRNA biomarker discovery in human biofluids. Performance parameters such as reproducibility, detection rate, and inter-platform correlation were used to evaluate each technology. MiRXES qRT-PCR and miRNA-Seq platforms had an almost perfect reproducibility between runs, calculating the Concordance Correlation Coefficient (CCC = 0.99), while the other three qRT-PCR platforms had moderate inter-run concordance (CCC > 0.9), and the NanoString platform had poor inter-run concordance (CCC = 0.82). The MiRXES qRT-PCR and NanoString platforms detected the highest and the lowest number of miRNAs above the LLOQ (lower limit of quantification) in serum samples, respectively. The authors concluded that the miRNA-Seq technology is preferable for discovery, while targeted qRT-PCR for subsequent validation of candidate extracellular miRNA biomarkers is recommended.

Finally, the droplet digital PCR (ddPCR) technique is becoming the gold standard in the application of liquid biopsy due to a number of advantages: (i) it allows an absolute quantification by means of sample partitioning and Poisson statistics (an internal/external normalization is thus not required); (ii) it has a superior precision and sensitivity in detecting low-abundant targets; (iii) it is less affected by PCR inhibitors [112,113,114,115]. However, ddPCR is less frequently used for c-miRNA measurements due also to a restricted multiplexing capacity, longer turnaround time for sample processing, and higher costs. In Table 3, a summary of the pros and cons of c-miRNA screening technologies is provided.

## 5. Discussion

Cancer biomarkers substantially change the way we manage lung cancer patients by improving disease stratification using intrinsic molecular characteristics, identification of therapeutic targets and monitoring molecular residual disease. However, the application of biomarkers for lung cancer early diagnosis is still limited by a lack of substantial trial-like research studies where the accuracy of proposed biomarkers is analyzed in real-world datasets. Previous studies highlighted pros and cons of different circulating biomarkers proposed for lung cancer detection and possible integration in the clinical routine (reviewed in Seijo et al. [116]). Circulating biomarkers can be very effective to inform clinical decision making in the management of indeterminate pulmonary nodules (IPNs) and in the management of diagnosed and resected lung cancer patients. Current management of IPNs is largely based on watchful waiting and may imply a risk of disease dissemination. Nodules found on annual LDCT screening, which are frequently very small in size and hamper current biopsy techniques, may benefit from an integrated risk model, which includes the different sources of information: clinical, imaging and biomarkers. This type of integrated risk model might also inform decisions regarding screening intervals, personalized follow-up of lung cancer patients, and prognostication.

Here, we made an update on recent developments in liquid biopsy-based biomarkers for lung cancer early diagnosis and proposed a roadmap for optimal biomarkers identification and development. A limit of this study is that we opted for a focused analysis on extensively validated biomarkers in large cohorts of samples including lung cancer screening studies rather than describing all circulating biomarkers proposed in the literature.

We have also brought to light the current limitations in biomarker research, which can be briefly summarized in: (i) poorly designed studies for biomarker discovery and validation; (ii) uncontrolled pre-analytic and analytic variabilities lacking standard operating procedures; (iii) frequent lack of validation studies using independent cohorts of samples collected from lung cancer screening studies; and (iv) somewhat sophisticated technologies for biomarker profiling that are hard to transfer to the clinical setting.

Biomarker research clearly offers substantial help in the characterization of at-risk population subgroups for screening selection and—more importantly—in the identification of disease precursors, predictive and prognostic factors before signs and symptoms of the disease appear. In particular, the analysis of liquid biopsies (i.e., plasma/serum) is emerging as promising for the quantification of biomarkers through also the use of lab-on-chip technologies, which would allow a rapid disease detection/monitoring and a biological characterization at the bedside [117,118]. Furthermore, genomic and proteomic breath tests besides airway epithelium signatures, are being trialed for early and non-invasive diagnosis of cancer and pulmonary disease, in particular for lung cancer and COPD [119,120]. Likewise, new emerging RNA-based biomarkers such as long non-coding RNA (lncRNA), circular RNA (circRNA) and platelets mRNAs have been described circulating in the blood with a potential for lung cancer early detection (Appendix A; Figure 2).

## 6. Conclusions

Thus far, all these multi-source biomarkers have never been combined into a coordinated and comprehensive workup for screening, diagnosis and treatment decision. The main barrier consists of difficulties in organizing worldwide large-scale studies with centralized resources for data/sample collection and processing following standard operating procedures. In addition, it is urgent to develop innovative approaches using big data and artificial intelligence (AI) analytics, such as machine learning, to improve both lung cancer early detection, personalized prevention strategies, and early treatments. We therefore look forward for these next-generation biomarkers in lung cancer screening programs to ameliorate early diagnosis, prognosis, and therapeutic response.

## Figures and Tables

**Figure 1 cancers-13-03919-f001:**
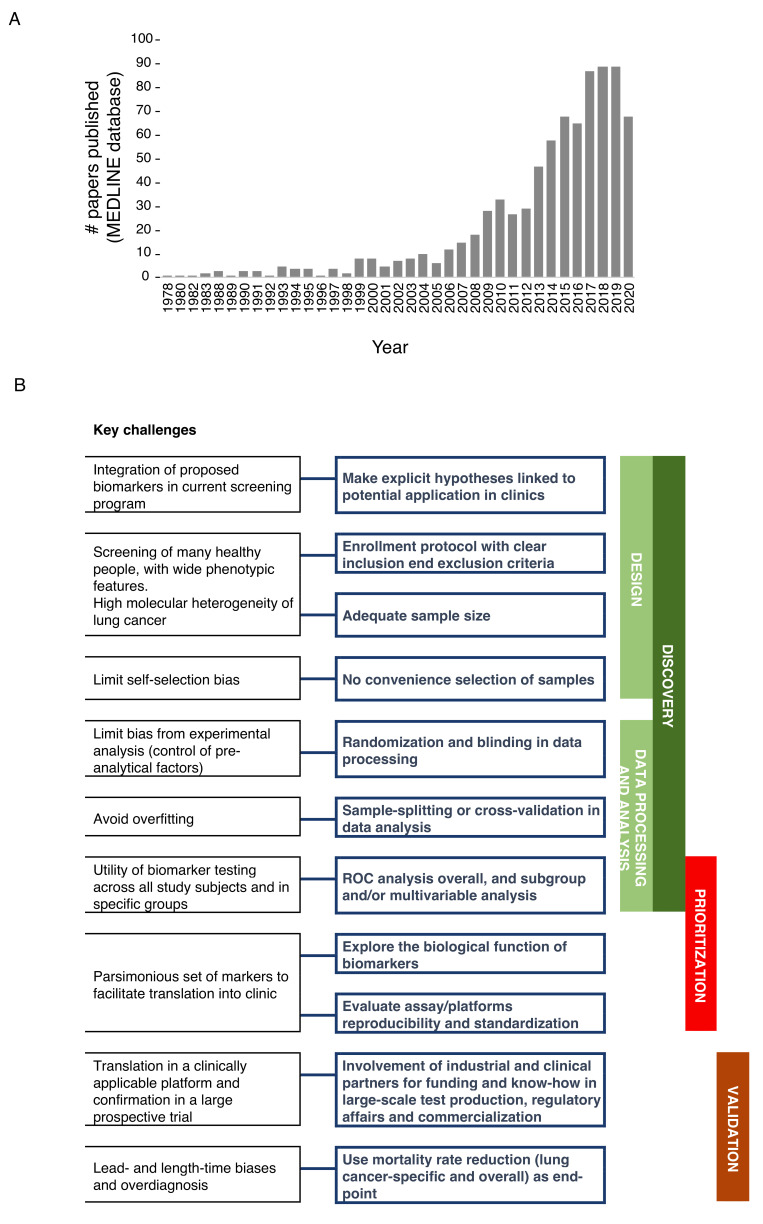
(**A**) Papers on lung cancer diagnostic biomarkers. PubMed free search engine which primarily accesses the MEDLINE database was interrogated (April 2021) by using ‘advanced search’ tool and with the following MESH terms: Lung neoplasms; Biomarkers; Diagnosis. (**B**) Schematic representation of best practice in biomarker development for early detection of lung cancer.

**Figure 2 cancers-13-03919-f002:**
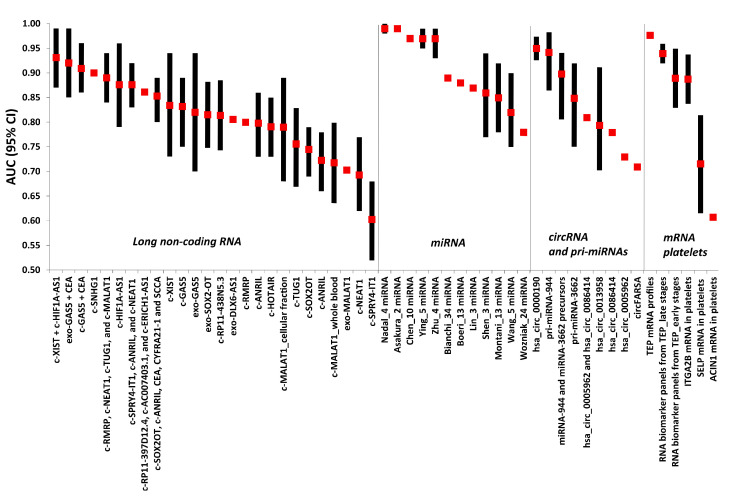
Forest plot showing the AUC and 95% confidence interval (when reported) for c-miRNA-based signatures (listed in Table 1) and other RNA-based biomarkers (listed in Appendix A). Red squares represent the AUC for each marker and black vertical bars extend from the lower limit to the upper limit of the 95% confidence interval (95% CI).

**Table 2 cancers-13-03919-t002:** Technical performance and clinical significance of circulating tumor cell (CTC) detection in early stage non-small cell lung cancer (NSCLC) patients and in screening programs.

Clinical Setting	*n* NSCLC Patients *	*n* Control Subjects *	CTC Enrichment and Detection Method	CTC Identification Criteria	Peripheral Blood Volume	*n* Target Cell-Positive Cases (Percentage)	Clinical Significance	Reference
preoperative	210 (191 stage I–III)	40 control subjects	EpCAM-based capture and expression of CK8–18-19 and CD45 (CellSearch) Size-based isolation by filtration through porous membranes (ISET, Rarecells) and staining with colorants for cytological samples	Round to oval morphology and CK+ CD45- for CellSearch Morphological features of malignancy for ISET	7 mL for CellSearch 10 mL for ISET	82 (39.3% stage I–III) by CellSearch 104 (49.5%; 49.7% stage I–III)0 control subjects by both technologies	EpCAM-positive selection is less sensitive than size-based isolation	Hofman V et al., Int J Cancer 2011 [82]
screening	0	245 cancer-free (168 COPD, 42 smokers, 35 non-smoking) subjects	Size-based isolation by filtration through porous membranes (ISET, Rarecells) Staining with colorants for cytological samples	Morphological features of malignancy	10 mL	5 COPD (3.0%) at first CT-scan→5 out of 5 confirmed diagnosis of lung cancer at subsequent scans	CTC detection anticipates lung cancer diagnosis by CT-scan screening (1 to 4 years)	Ilie M et al., Plos One 2014 [74]
screening	15 (advanced lung ca.)	32 GGO 19 no GGO	Antibody-based capture of EpCAM+ cells (GILUPI CellCollector, GILUPI) _a) EpCAM/CK and CD45 expression by immunofluorescence, and morphological features by imaging analysis(b) Cancer-related gene panel mutations by NGS	EpCAM+/CK+ CD45- and mutated cancer genes	Estimated 1.5–3 liters	11 patients (73.3%) 5 GGO (15.63%) 0 no GGO	CTC can be detected in subjects with preneoplastic nodules and can differentiate GGO from no GGO	He Y et al., Sci Rep 2017 [83]
screening	0	High-risk individuals (smoking habits, age, chronic infections, PSA level) 3888	Microfluidics for flow rate-, surface interaction-, plasticity-, and elasticity-based cell separation (IsoPic, iCellate) Pan-CK and CD45 expression by immunofluorescence	CK+ CD45−	7.5 mL	107 (3.2%) patients	Detection frequency compatible with screening-detected lung cancer rate; follow-up needed to validate results	Castro J et al., Dis Markers 2018 [84]
screening	29 treatment-naïve (stage I–IV)	31 high-risk *w*/ or *w*/o benign nodules 20 control subjects	Size-based isolation by filtration through filters with a syringe pump (CellSieve Creatv MicroTech) CK8/18/19, EpCAM and CD45 expression assessed by Immunofluorescence	CK+/EpCAM+ CD45− (single cells or cluster of ≥2 cells)	7.5 mL	Single CTC: 29 patients (100%) 18 high-risk (58.1%) 0 control subjects CTC cluster: 12 patients (41.4%) 0 high-risk or control subjects	High detection rate of single target cells, good specificity of clusters	Manjunath Y et al., Lung Cancer 2019 [85]
screening	115 (97 stage I–III)	87 long-term smokers 20 healthy controls	Size-based isolation by filtration through filters with a syringe pump (CellSieve, Creatv MicroTech) CK8/18/19, EpCAM, CD14 and CD45 expression assessed by immunofluorescence	Cell diameter ≥30 μm, CK+/EpCAM+ CD14+/ CD45+	7.5 mL	88 patients (86.5%): 38 (65.5%) stage I, 13 (72.2%) stage II, 19 (90.5%) stage III 6 long-term smokers (6.9%)0 healthy controls	High specificity and sensitivity of tumor-macrophage-hybrid cells	Manjunath Y et al., JTO 2020 [86]
screening	19 (Stage I–IV screening-detected)	592 LDCT-screened lung cancer-free heavy smokers	Size-based isolation by filtration through porous membranes (ISET, Rarecells) Staining with colorants for cytological samples	Morphological features of malignancy	10 mL	22 control cases (3.7%) 5 patients (26.3%)	CTC detection rate not sufficient for application in screening programs	Marquette CH et al., Lancet Respir Med 2020 [79]
preoperative	34 (non-metastatic)	20 lung cancer-free 10 benign lung nodules	Antibody-based capture of EpCAM+ cells (GILUPI CellCollector, GILUPI) (a) Cytokeratin CK7/19/panCK, PD-L1 and CD45 expression by immunofluorescence(b) DNA CNV by NGS	CK+ CD45− and DNA CNV	Estimated 1.5–3 liters [83]	18 patients (52.9%) 1 control case (3.3%)	Technical approach able to validate CTC authenticity	Duan G-C et al. OncoTargets and Therapy 2020 [87]
screening	107 (67% stage I–II)	100 lung cancer-free individuals	Ficoll density gradient collection of PBMC 4-color FISH with probes at 10q22.3/CEP10 and 3p22.1/3q29	Polysomy in at least two fluorescence channels	10 mL	95 patients (88.8%) 0 control subjects	Genetically abnormal circulating cells can be detected with high accuracy	Katz DL et al., Cancer Cytopathol 2020 [80]

* Evaluable for CTC analysis. Abbreviations: CK, cytokeratin; ISET, Isolation by Size of Epithelial/Tumor cells; COPD, chronic obstructive pulmonary disease; GGO, ground-glass opacity nodule; PSA, prostate-specific antigen; LDCT, low-dose computed tomography; NGS, next generation sequencing; CNV, copy-number variation; FISH, fluorescence in situ hybridization; PBMC, peripheral blood mononuclear cells.

**Table 3 cancers-13-03919-t003:** miRNA platforms comparison.

Method	Platform (Vendor)	Turnaround Time	Costs Per Sample	Panel Content (Human miRNA)	Reproducibility	SE	SP	ACC
qRT-PCR	miScript (Qiagen)	+++	$$	1066	Medium [105,108]	Medium [105]	Medium [105]	High [105]
	miRCURY Exiqon (Qiagen)	+++	$$	752	High [105] -Medium [108]	Medium [105,108]	High [105] -Medium [105]	High [105]
	TaqMan Cards preAMP (Life Technologies) ^a^	+++	$$	754	Medium [108]-Low	High [94]-Medium [108]	High [105] -Medium [105]	High [105]
	TaqMan OpenArray (Life Technologies) ^a^	+	$	754	Low [105]	Medium [105]	High [105]	High [105]
	SmartChip (WaferGen)	+++	$$	1036	High [105]	Low [105]	Medium [105] -Low [105]	Low [105]
	qScript (Quanta BioSciences)	+++	$$	489	High [105]	Medium [105]	High [105] -Medium [105]	High [105]
	miRXES ID3EAL (miRXES)	+++	$$	560	High [108]	High [108]	NA	NA
GeneChip miRNA arrays	microarray (Affymetrix)	++	$	Up-to-date content from miRBase 20	Medium [105]	NA	High [105]- Low [105]	Low [105]
	microarray (Agilent)	++	$	Up-to-date content from miRBase 21	High [105]	Low [105]	High [105] -Medium [105]	Low [105]
nCounter platform	nCounter (NanoString)	+	$	800	Medium [105]-Low [108]	Low [107,108]	High [105] -Medium [107]	Low [105]
sRNA-Seq (miRNA-seq)	TruSeq (Illumina)	++	$	Up to 2693 ^b^ (miRBase 22)	High [105,107,108]	High [107,108]-Medium [105]	High [105,107]	Medium [105]
	Ion Torrent (Life Technologies)	++	$	Up to 2693 ^b^ (miRBase 22)	Medium [105]	Medium [105]	Low [105]	Medium [105]
HTG EdgeSeq	HTG EdgeSeq (HTG Molecular Diagnostics) plus Illumina or Thermo Fisher Ion Torrent sequencers	+	$$	2083	High [107]	High [107]	High [107]	NA
Standard flow cytometer	FirePlex (Abcam)	+	$$	up to 65 miRNAs per well	Low [107]	Medium [107]	Low [107]	NA

Platform comparison. ^a^ Standard TaqMan MicroRNA Assays use a target-specific stem-loop reverse transcription primer; ^b^ Number of mature microRNAs in miRBase release 22; NA = not analyzed [105,107,108]. From $ to $$, qualitative scale of costs for sample processing. From + to +++, qualitative scale of turnaround time for sample processing.

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
