# Peer review of "Biomarkers and Lung Cancer Early Detection: State of the Art"

_cancers, 2021, doi:10.3390/cancers13153919_

Round 1
Reviewer 1 Report
I have no special comments to the authors
Author Response
please see the attached rebuttal letter

Reviewer 2 Report
Certainly a very informative and complete review. However I have several concerns and/or provisos.
1. Abstract is not informative.
2. There are no clear conclusions.
3.There are no statements on the limitations of the review.
4. I cannot see what could be if any the method to discern benign lung nodes from lung cancer.
5. I cannot see ROC AUC for the studies mentioned.
6. In addition to the Tables, a sub-group analysis Forest plot including AUCs could be of great guidance.
7. I cannot see studies on platelets. Se Best et al. RNA-Seq of tumor-educated platelets enables blood-based pan-cancer, multiclass, and molecular pathway cancer diagnostics. Cancer Cell 2015, Best et al. Swarm intelligence-enhanced detection of non-small cell lung cancer using tumor-educated platelets. Cancer Cell 2017.
Author Response
please see the attached rebuttal letter

Reviewer 3 Report
In my opinion the idea of the evaluated review type paper is on time. However, I have few comments addressed to authors:
- The numerous information that were described is already well-known and there is lack of necessity to repeat it again. It is better to touch most recent findings. Authors, should focus on the new markers such as lncRNAs, circRNAs and pri-miRNAs, that are still not well known.
- The most important clusters of markers should be summarized in tables. Unfortunately, there is a deficit of specific markers and described too generally.
- As mentioned above, the more precise screening of the latest literature reports should be made.
- Some discussion on the pros and cons of molecular markers for LC should be introduced to manuscript body.
Author Response
please see the attached rebuttal letter

Round 2
Reviewer 2 Report
I have no additional comments to add.
Reviewer 3 Report
Authors corrected paper according to reviewer's comments.